# Effectiveness of BNT162b2 and CoronaVac in children and adolescents against SARS-CoV-2 infection during Omicron BA.2 wave in Hong Kong

Daniel Leung [1,2], Jaime S. Rosa Duque [1,2], Ka Man Yip [1], Hung Kwan So [1], Wilfred H. S. Wong [1,3✉] & Yu Lung Lau [1,3✉]

**Abstract**

**Background** The SARS-CoV-2 Omicron BA.2 subvariant replaced BA.1 globally in early 2022, and caused an unprecedented tsunami of cases in Hong Kong, resulting in the collapse of elimination strategy. Vaccine effectiveness (VE) of BNT162b2 and CoronaVac against BA.2 is unclear.

**Methods** We utilize an ecological design incorporating population-level vaccine coverage statistics and territory-wide case-level SARS-CoV-2 infection surveillance data, and investigate the VE against infection during the Omicron BA.2 wave between January 1 to April 19, 2022, in Hong Kong for children and adolescents.

**Results** We estimate VE to be 33.0% for 1 dose of BNT162b2 in children aged 5–11 and 40.8% for 2 doses of CoronaVac in children aged 3–11. We also estimate 54.9% VE for 2 doses of BNT162b2, and 55.0% VE for 2 doses of CoronaVac in adolescents aged 12–18.

**Conclusions** Our findings support partly preserved VE against infection by variants of concerns for children and adolescents in settings with extremely low levels of prior SARS-CoV-2 circulation.

**Plain language summary**

COVID-19 vaccines BNT162b2 and CoronaVac are widely used globally in children and adolescents, yet their effectiveness against Omicron BA.2 variant—a version of the SARS-CoV-2 virus that became predominant in early 2022—is not fully known. In Hong Kong, we combine population-level vaccine coverage statistics and territory-wide SARS-CoV-2 infection data, and estimate their effectiveness against SARS-CoV-2 infection in children and adolescents during a major Omicron BA.2 outbreak. We find moderate vaccine effectiveness after 2 doses of BNT162b2 or CoronaVac, supporting the use of either vaccine in children and adolescents during a rapidly evolving pandemic.

[1] Department of Paediatrics and Adolescent Medicine, The University of Hong Kong, Hong Kong, China. [2] These authors contributed equally: Daniel Leung, Jaime S. Rosa Duque. [3] These authors jointly supervised this work: Wilfred H. S. Wong, Yu Lung Lau. ✉email: whswong@hku.hk; lauylung@hku.hk

After 2 years of experience with SARS-CoV-2, coronavirus disease 2019 (COVID-19) is now understood to be usually milder in healthy children than adults, but still leads to multisystem inflammatory syndrome in children and necessitate hospitalization in some[1,2]. BNT162b2 (BioNTech with Pfizer or Fosun Pharma) and CoronaVac (Sinovac) are the two COVID-19 vaccines with the highest number of doses distributed internationally[3], and both have been authorized for paediatric populations in various countries and regions, including Hong Kong. In April 2021, Pfizer and BioNTech requested regulatory agencies worldwide to lower the minimum age of BNT162b2 to 12, based on non-inferior neutralization and observed 100% efficacy against COVID-19[4]. Following the release of positive interim results for BNT162b2 in children aged 5–11 in September 2021, with non-inferior neutralization and 90.7% efficacy found[5], children aged 5–11 were also allowed to receive the 10-mcg paediatric formula of BNT162b2. For CoronaVac, vaccine efficacy was established in adults[6]. A Phase 2b study in mainland China showed positive immunogenicity results of the adult dose of CoronaVac in children and adolescents aged 3–17, with no efficacy data[7].

Post-licensure monitoring of vaccine effectiveness (VE) in children and adolescents is important as children and adolescents may have a different immune response to vaccines from adults[8], and immunobridging studies supporting the paediatric use of COVID-19 vaccines may lack efficacy data. In late 2021, the Omicron BA.1 variant was first detected in southern Africa, associated with dramatic neutralization escape that may justify the classification of the lineage as a novel serotype of SARS-CoV-2[9,10], and it caused a major global surge. Several VE studies performed during BA.1 predominance have been published[11], including in children and adolescents[12–21], mostly showing a reduced VE against symptomatic COVID-19 but preserved VE against severe outcomes. Classified under the same lineage is an antigenically distinct subvariant BA.2[22,23], which overtook BA.1 in early March 2022 globally on GISAID[24], and led to the peak of the first Omicron wave in some countries and regions including Denmark and Hong Kong[25]. Globally, the vaccine effectiveness against BA.2 infection in children and adolescents aged 3–18 remains unclear. Moreover, most VE studies have been conducted in countries with high circulation of pre-Omicron variants, which may lead to underestimating VE as history of previous COVID-19 in participants could not be fully ascertained.

Hong Kong has persisted in an elimination strategy by adopting various nonpharmaceutical interventions against SARS-CoV-2 from the beginning of the pandemic[26–28], with no more than 13,000 confirmed cases prior to 2022, in a population of 7.5 million. In January 2022, Hong Kong reported the first major local cluster of Omicron BA.2.2 in a densely populated public housing estate[25], which was linked to a traveler from Pakistan, who contracted BA.2.2 in a quarantine hotel from two infected travelers from Nepal, proven by genomic sequencing[29]. An unprecedented tsunami of SARS-CoV-2 infections, predominated by BA.2.2, quickly followed, unabated by nonpharmaceutical interventions.

In this study, we analyzed VE against infection conferred by BNT162b2 in children and adolescents aged 5–18 years and CoronaVac in children and adolescents aged 3–18 years residing in Hong Kong between 1 January 2022 and 19 April 2022 with a retrospective ecological design, leveraging territory-wide population-level vaccine coverage statistics as well as case-based infection surveillance data from the Department of Health in Hong Kong. Deidentified data of infected cases aged 3–18 were extracted from the Department of Health database, including reporting date, age, gender, and vaccination dates and brands. Cases with heterologous prime-boost or vaccine brands other than BNT162b2 and CoronaVac were excluded. We obtained daily territory-wide vaccine coverage statistics from the Department of Health, which were stratified by age (3–11 years and 12–18 years) and gender. We calculated the incidence rates in each category defined by age, gender, and vaccination status on each day during the study period, and estimated the incidence rate ratios (IRR) against the unvaccinated reference group by negative binomial regression for the daily counts of cases, adjusted for gender and calendar day for the 2 vaccine brands in each age group, as there were differences in vaccination coverage and infection incidence, and used the logarithm of persons-at-risk as the offset variable. We estimated VE as $(1-IRR)\cdot100\%$, with a two-sided 95% confidence interval (CI). VE estimates for dose 2 of BNT162b2 in children and dose 3 of either vaccine in adolescents were not estimated as rollout took place late in the study period. In addition, based on the VE point estimates, we studied the direct impact of the vaccination programme during the study period for the 2 age groups. The expected number of cases in the absence of the vaccination programme is estimated by $case_0 + case_{BNT1}/(1 - VE_{BNT1}) + case_{BNT2/3}/(1 - VE_{BNT2}) + case_{CoV1}/(1 - VE_{CoV1}) + case_{CoV2/3}/(1 - VE_{CoV2})$, where $case_{0/BNT/CoV1/2/3}$ is the number of observed unvaccinated cases or cases with 1/2/3 doses of BNT162b2 or CoronaVac and $VE_{BNT/CoV1/2}$ is the VE of 1 or 2 doses of BNT162b2 or CoronaVac[30]. The numbers of cases averted are given by deducting the expected numbers of cases with the observed. Additional details are available in Methods. We estimate VE to be 33.0% for 1 dose of BNT162b2 and 40.8% for 2 doses of CoronaVac in children. We also find 54.9% VE for 2 doses of BNT162b2, and 55.0% VE for 2 doses of CoronaVac in adolescents.

## Methods

**Study design**. We analyzed VE against infection conferred by BNT162b2 in children aged 5–18 years and CoronaVac in children aged 3–18 years residing in Hong Kong during a major Omicron BA.2.2 outbreak with a retrospective ecological design. The study is approved by the University of Hong Kong Institutional Review Board (UW 22–217). We utilized territory-wide population-level vaccine coverage statistics as well as infection case data from the Department of Health, Hong Kong Special Administrative Region from 1 January 2022 to 19 April 2022, during which the city had its first major outbreak of COVID-19. Hong Kong has persisted in an elimination strategy from the beginning, with no more than 13,000 confirmed cases in the first 2 years of the pandemic, prior to the study period. Tough border control measures, including place-specific flight suspension, extended quarantine for most inbound travellers in designated quarantine hotels, and circuit-breaker mechanism for international flights had prevented the introduction of pre-Omicron variants into community circulation in Hong Kong. Masking in all public places has been mandatory in Hong Kong since July 2020, and various other social distancing measures, such as suspension or reduced operational hours of restaurants, bars, gyms, and parks, had also been put in place intermittently. Before the roll-out of COVID-19 vaccine, schooling was suspended or online only, with only about 3 months of face-to-face teaching in 2020[31]. In January 2022, following the BA.2 outbreak, Hong Kong promptly suspended face-to-face teaching, which was not fully resumed until after the end of the study period.

**Testing and reporting of SARS-CoV-2 infections in Hong Kong**. In Hong Kong, in accordance with Prevention and Control of Disease Regulation (Cap. 599), persons suspected by medical practitioners to be infected, or persons at high risk of infection, e.g., close contacts, those residing in buildings with confirmed

cases, those who have visited restaurants or premises known to be visited by infected cases, were subjected to compulsory polymerase chain reaction testing. Such compulsory testing notices were regularly enforced by the police, and failure to comply may result in prosecution. Testing may be done via submission of deep throat saliva (or stool for young children) samples, nasopharyngeal swabbing, or combined nasal and throat swabbing in community testing centres, or mobile testing stations within Government-designated high-risk areas known as 'restriction areas' which were cordoned off overnight. Positive test results were reported to the Department of Health. There could be delays and backlogs during testing or reporting due to sudden high volume of specimens or reports. From March 2022, rapid antigen self-tests have become widely available, and positive results could be reported by citizens online to the Department of Health (https://www.chp.gov.hk/ratp) and are included in the case count. In this analysis, deidentified infection data of patients aged 3–18 years were extracted from the Department of Health database, which included the reporting date, demographics (age and gender) and vaccination status (unvaccinated or vaccinated; number of doses, brand, and date) of each case. Vaccination status was extracted from the Department of Health database as all COVID-19 vaccines in Hong Kong were administered and recorded by the Government. Cases with heterologous prime-boost or vaccine brands other than BNT162b2 and CoronaVac were excluded. The number of cases in the study population on each day during the study period by vaccination status is presented in Fig. 1c, d for the two age groups.

**Vaccination programme and coverage in Hong Kong.** COVID-19 vaccination is coordinated by the Government in Hong Kong, with only the adult formula of BNT162b2 and CoronaVac available. The minimum age for receiving BNT162b2 and CoronaVac was lowered in a step-wise manner for adolescents and children in Hong Kong. Vaccination with 2 doses BNT162b2, 21 days apart, in adolescents aged 16–18 was recommended by the Joint Scientific Committees (JSC) on Vaccine Preventable Diseases and on Emerging and Zoonotic Diseases under the Department of Health, joined by the Chief Executive's Expert Advisory Panel in January 2021, and vaccines became available in late February 2021. In June 2021, the JSC also recommended the use of 2-dose BNT162b2 in adolescents aged 12–15; the second dose of BNT162b2 in adolescents aged 12–17 years was suspended in September 2021 due to higher risk of post-mRNA vaccine myopericarditis in younger population. In November 2021, 2 doses of CoronaVac, 28 days apart, were recommended for adolescents aged 12–17. In view of likely local Omicron outbreak, the second dose of BNT162b2 was reinstated for adolescents aged 12–17 in December 2021, at a 12-week interval, subsequently shortened to an 8-week interval. The minimum age for CoronaVac was lowered to 5 in January 2022 and to 3 in the next month, at the same dosage for all ages. A third dose was recommended to adolescents aged 12–17 who received 2 doses of CoronaVac at least 3 months before, and for those received 2 doses of BNT162b2 at least 5 months before, respectively, in February 2022. While Hong Kong did not receive the paediatric formula for BNT162b2, 2 doses of one-third fractional dose was

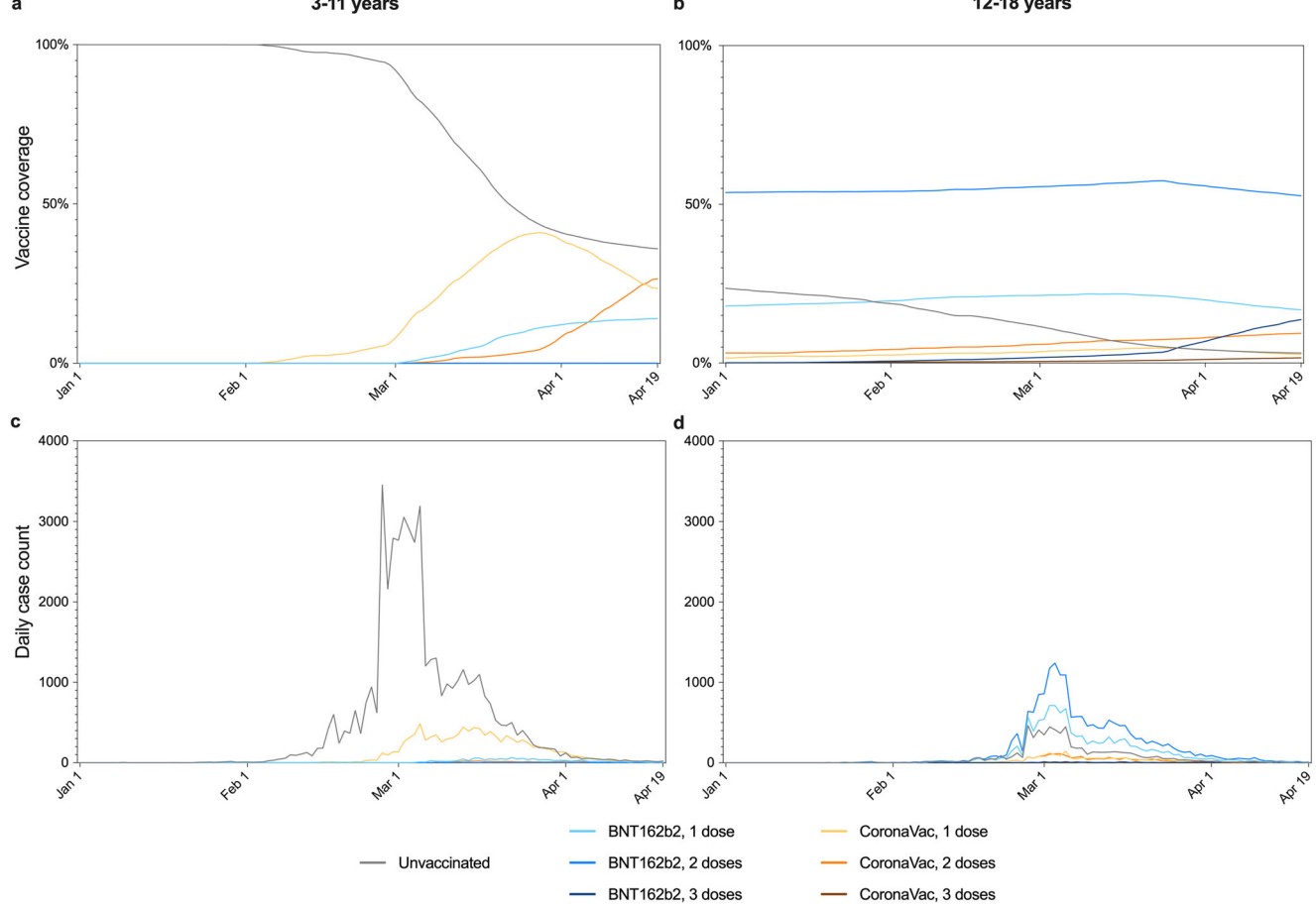

**Fig. 1 Vaccine coverage and daily number of cases in children (ages 3–11) and adolescents (ages 12–18) in Hong Kong during the study period.** Panels **a**, **b** show the vaccine coverage in children and adolescents in Hong Kong respectively, by vaccine brand and doses, while **c**, **d** show the daily number of cases by vaccine brand and doses in children and adolescents, respectively.

authorized in February 2022 for children aged 5–11 years, given at least 8 weeks apart. The third dose was recommended for children aged 3–11 years who received CoronaVac in March 2022, at a 3-month interval from the second dose. Starting in February 2022, Hong Kong implemented the first stage of the Vaccine Pass policy, whereby all persons aged 12 years or above are required to have received at least 1 dose of COVID-19 vaccine to enter restaurants, bars, gyms, theme parks, and other specified premises.

We obtained daily territory-wide vaccine coverage statistics from the Department of Health, which were stratified by age (3–11 years and 12–18 years) and gender, during the study period. The number of unvaccinated children was estimated by deducting the number of vaccinated children from the population size data from the local census at the end of 2021.

**VE estimation**. We classified vaccination statuses (unvaccinated and 1, 2, and 3 doses of BNT162b2 and CoronaVac) in the infected and in the population by the date of vaccination with a 14-day lag. Daily counts of vaccines for each vaccination status in the population were estimated on the assumption of homologous vaccination as that was the recommended practice. The population at risk in each category on each day, defined by vaccine status, age group (3–11 years and 12–18 years), and gender, was calculated by subtracting daily count of vaccinees with cumulated infections in that category. We calculated the incidence rates in each category on each day during the study period, and estimated the incidence rate ratios (IRR) against the unvaccinated reference group by negative binomial regression for the daily counts of cases with the logarithm of persons-at-risk as an offset variable, adjusted for gender (boys or girls) and calendar day ("1" for the first day of the study period, and so on) for the 2 vaccine brands in each age group, as vaccination coverage and incidence differed by gender and age group. We estimated VE as (1-IRR)·100%, with a two-sided 95% confidence interval (CI). Sample size was not estimated as this is a territory-wide study. Impact of pre-Omicron COVID-19 history and exposure risks such as masking or living environment on VE were not assessed as incidence of pre-Omicron COVID-19 was very low in Hong Kong and such information was not available. The expected number of cases in the absence of the vaccination programme is estimated by $case_0 + case_{BNT1}/(1 - VE_{BNT1}) + case_{BNT2/3}/(1 - VE_{BNT2}) + case_{CoV1}/(1 - VE_{CoV1}) + case_{CoV2/3}/(1 - VE_{CoV2})$, where $case_{0/BNT/CoV1/2/3}$ is the number of observed unvaccinated cases or cases with 1/2/3 doses of BNT162b2 or CoronaVac and $VE_{BNT/CoV1/2}$ is the VE of 1 or 2 doses of BNT162b2 or CoronaVac[30]. 95% CIs of expected cases are estimated using bootstrap based on 1000 resamples. The numbers of cases averted are given by deducting the expected numbers of cases with the observed. Statistical analysis was performed using R (Version 4.0.3).

**Reporting Summary**. Further information on research design is available in the Nature Portfolio Reporting Summary linked to this article.

## Results

**Vaccine effectiveness**. At the end of the study period from 1 January 2022 to 19 April 2022, 64.0% of 506100 persons aged 3–11 and 96.9% of 447300 persons aged 12–18 in the population have been vaccinated with at least 1 dose (Table 1; Fig. 1a, b). Among vaccinees aged 3–11 and 12–18, 22.0% and 85.9%, respectively, received BNT162b2. In the study period, 100274 cases were reported to the Department of Health in Hong Kong in ages 3–18, including 137 cases (0.14%) with heterologous prime-boost or vaccine brands other than BNT162b2 and CoronaVac, which were excluded. 59969 and 40168 cases were included in those aged 3–11 and 12–18 respectively (Table 1; Fig. 1c, d). A total of 50472 cases were tested by polymerase chain reaction, while 49665 cases were self-reported rapid antigen tests. We estimated VE against infection after 1 dose of BNT162b2 and 1 and 2 doses of CoronaVac in those aged 3–11, and after 1 and 2 doses of either vaccine brand in those aged 12–18 (Table 2). We calculated a VE of 33.0% (95% CI 3.0–53.3%) conferred by 1 dose of BNT162b2 in those aged 5–11. We were unable to estimate the VE for 2 doses of BNT162b2 in those aged 5–11 as the minimum age for BNT162b2 was lowered to 5 in February 2022, and the second dose was recommended to be given 8 weeks after the first. While the VE estimate for 1 dose of CoronaVac was −14.7% (−54.7–14.6%), the VE estimate for 2 doses of CoronaVac was

**Table 1 Characteristics of the total population and reported cases of the population aged 3-11 and 12-18 in Hong Kong.**

| Characteristic | | 3–11 years | | | | 12–18 years | | | |
| --- | --- | --- | --- | --- | --- | --- | --- | --- | --- |
| | | Female | | Male | | Female | | Male | |
| | | N | Col % | N | Col % | N | Col % | N | Col % |
| Population size | | 245700 | | 260400 | | 218140 | | 229160 | |
| Vaccination total | | 156141 | 63.5% | 167986 | 64.5% | 210273 | 96.4% | 223033 | 97.3% |
| Vaccine brand | Dose | | | | | | | | |
| BNT162b2 | 1 | 34576 | 14.1% | 36403 | 14.0% | 34563 | 15.8% | 40423 | 17.6% |
| | 2 | 121 | 0.0% | 107 | 0.0% | 116063 | 53.2% | 119718 | 52.2% |
| | 3 | 0 | 0.0% | 2 | 0.0% | 31135 | 14.3% | 30102 | 13.1% |
| CoronaVac | 1 | 56969 | 23.2% | 61709 | 23.7% | 5500 | 2.5% | 6638 | 2.9% |
| | 2 | 64456 | 26.2% | 69744 | 26.8% | 19494 | 8.9% | 22384 | 9.8% |
| | 3 | 19 | 0.0% | 21 | 0.0% | 3518 | 1.6% | 3768 | 1.6% |
| Unvaccinated | 0 | 89559 | 36.5% | 92414 | 35.5% | 7867 | 3.6% | 6127 | 2.7% |
| Infection total | | 27804 | 11.3% | 32165 | 12.4% | 18776 | 8.6% | 21392 | 9.3% |
| Vaccine brand | Dose | | | | | | | | |
| BNT162b2 | 1 | 576 | 2.1% | 663 | 2.1% | 4949 | 26.4% | 6069 | 28.4% |
| | 2 | 0 | 0.0% | 0 | 0.0% | 9047 | 48.2% | 9230 | 43.1% |
| | 3 | 0 | 0.0% | 0 | 0.0% | 131 | 0.7% | 131 | 0.6% |
| CoronaVac | 1 | 4824 | 17.4% | 5614 | 17.5% | 964 | 5.1% | 1090 | 5.1% |
| | 2 | 240 | 0.9% | 293 | 0.9% | 880 | 4.7% | 1065 | 5.0% |
| | 3 | 0 | 0.0% | 1 | 0.0% | 26 | 0.1% | 21 | 0.1% |
| Unvaccinated | 0 | 22164 | 79.7% | 25594 | 79.6% | 2779 | 14.8% | 3786 | 17.7% |

**Table 2 Vaccine effectiveness against infection, at least 14 days after each dose in children (ages 3–11) and adolescents (ages 12–18).**

| | Number of infections | Mean days elapsed from last dose (SD) | VE estimate | 95% CI | *p*-value |
|---|---|---|---|---|---|
| Children (ages 3–11) | | | | | |
| BNT162b2, 1 dose | 1239 | 24 (9) | 33.0% | 3.0–53.3 | 0.03 |
| CoronaVac, 1 dose | 10438 | 25 (7) | −14.7% | −54.7–14.6 | 0.36 |
| CoronaVac, 2 doses | 533 | 25 (20) | 40.8% | 12.8–59.5 | 0.008 |
| Unvaccinated | 47758 | ref | ref | ref | ref |
| Adolescents (ages 12–18) | | | | | |
| BNT162b2, 1 dose | 11018 | 122 (62) | 26.1% | −0.3–45.6 | 0.054 |
| CoronaVac, 1 dose | 2054 | 31 (13) | 21.5% | −7.7–42.7 | 0.14 |
| BNT162b2, 2 doses | 18277 | 189 (51) | 54.9% | 38.9–66.8 | <0.0001 |
| CoronaVac, 2 doses | 1945 | 64 (54) | 55.0% | 38.2–67.2 | <0.0001 |
| Unvaccinated | 6565 | ref | ref | ref | ref |

**Fig. 2 Daily expected number of cases in the absence of the vaccination programme during the BA.2 outbreak in children and adolescents in Hong Kong.** Panel **a** refers to 3–11 years, while Panel **b** refers to 12–18 years. 95% confidence interval for expected number of cases is highlighted in pink.

40.8% (12.8–59.5%) in those aged 3–11. In those aged 12–18, we found a VE of 26.1% (−0.3–45.6%) for 1 dose of BNT162b2, and 54.9% (38.9–66.8%) for 2 doses of BNT162b2. Meanwhile, VE estimates were 21.5% (−7.7–42.7%) for 1 dose of CoronaVac, and 54.9% (38.2–67.2%) for 2 doses of CoronaVac. VE estimates for dose 3 in adolescents were not estimated as rollout was late in the study period.

**Cases averted by the vaccination programme**. Using these VE estimates, we calculated the expected numbers of cases in the absence of the vaccination programme by dividing the observed number of cases for each age group and vaccination status with 1-VE, with the expected and observed daily day counts depicted in Fig. 2. We found an expected 60947 and 69674 number of cases in ages 3–11 and 12–18 during the study period, meaning 979 and 29506 cases were averted in ages 3–11 and 12–18 by the vaccination programme during the BA.2 outbreak in Hong Kong.

## Discussion

While Omicron BA.2 is known to evade neutralizing antibody responses, we documented a moderate VE against infection in children and adolescents after 2 doses in Hong Kong with both BNT162b2 and CoronaVac.

In adolescents who received 2 doses of BNT162b2, estimates from some studies have shown waning VE to varying degrees at more than 5 months after vaccination, from 62% estimated in a prospective cohort study by Fowlkes et al.[19], to near-zero VE

estimates found by Klein et al with a test-negative design in patients with emergency department or urgent care encounters[18]. In our study, infections occurred on average 189 days after 2 doses of BNT162b2 in adolescents, and the minimum age for BNT162b2 has been lowered to 12 since June 2021, i.e., 6 months before the study period in Hong Kong. In another study, our group has found that neutralizing antibodies against Omicron BA.1 were detectable in 96% of adolescents 6 months after 2 doses of BNT162b2[32]. These results suggest durable effectiveness against infection, even against variants of concerns, in adolescents vaccinated with BNT162b2, possibly more evident in a setting with low prior circulation such as Hong Kong. Studies have also shown high VE against BA.1 infections shortly after 3 doses in adolescents. Fleming-Dutra et al estimated a 71% VE against symptomatic COVID-19 between 2 to 6.5 weeks after receiving 3 doses of BNT162b2 in adolescents aged 12–15, using a test-negative design[12]. Klein et al found an 81% VE against emergency department or urgent care encounters 1 week after 3 doses of BNT162b2 in adolescents aged 16–17[18]. While we were only able to estimate VE conferred by 1 dose of BNT162b2 in children, we hypothesize VE against infection after 2 doses to be similar to that in adolescents, and we note point estimates of ~45–65% VE against infection or mild outcomes in several studies in children who received 2 doses of BNT162b2[12,16,18,20].

VE estimates for inactivated COVID-19 vaccines are scarce, especially against Omicron and in children and adolescents. During the BA.1 wave in Brazil, Florentino et al. estimated a 42% VE against symptomatic COVID-19 in children aged 6–11 who

received 2 doses of CoronaVac with a test-negative design[21]. In addition, Jara et al found 38% VE conferred by 2 doses of CoronaVac against symptomatic Omicron COVID-19 in a total population cohort of Chilean children aged 3–5[17]. Meanwhile, we found 41 and 55% VE in children aged 3–11 and adolescents aged 12–18 who received 2 doses of CoronaVac, respectively. Inactivated COVID-19 vaccines, in general, elicit poorer neutralizing antibody responses than mRNA vaccines, including in adolescents[8], and against variants of concerns[22,33], yet cellular responses against non-Spike structural proteins may explain preserved VE against variants of concerns in children and adolescents[34–36].

It is also interesting to understand VE estimates in our study together with those in adults. Leveraging a territory-wide hospital dataset, McNenamin et al estimated VE against mild or moderate COVID-19 during Hong Kong's BA.2 wave in adults[37], and found 31 and 18% VE estimates in those aged 20–59 who received 2 doses of BNT162b2 and CoronaVac respectively, and no VE in those aged 60 years or above. We suggest this may be explained by superior immune responses to vaccination in children and adolescents, such as the superior antibody responses found in phase 2/3 trials of both BNT162b2 and CoronaVac as well as other studies[4,5,7,8,38]. Yet, as COVID-19 vaccination was rolled out earlier to adults than to children and adolescents, the difference may also be partially explained by waning.

It is important to understand that different VE studies have different limitations and were performed in different settings, and therefore each VE study, including this one, should be interpreted with all available literature[39]. Our findings should be interpreted with the following limitations in mind. Importantly, we only investigated VE against infection, which are likely lower than VE against severe outcomes. Observational designs are subject to bias, e.g., care-seeking bias or other confounders. We are unable to adjust for individual level covariates with ecological design, e.g., prior COVID-19 and comorbidities, yet prior COVID-19 is very rare in Hong Kong, and comorbidities increasing the risk of infection or reducing VE are generally rare in children. We also could not account for time since vaccination to evaluate waning in this analysis. Our findings reveal VE in a setting with very low prior circulation, and are important to regions with zero-COVID policy such as mainland China, yet may not apply to regions with previously high circulation. We categorized person-time under 14 days of vaccination under the previous dose, or unvaccinated for 14 days after dose 1. This may underestimate the VE of the previous dose if there was exposure at a crowded community vaccination centre, yet masking is mandatory in public settings in Hong Kong. 76% adolescents have received at least one dose at the beginning of the study period, which may increase likelihood for bias in that those unvaccinated may have a different level of risk than the vaccinated. However, we believe care-seeking behaviour and personal hygiene practice between those unvaccinated and vaccinated to be similar as the authors observed universal compliance with masking in Hong Kong regardless of vaccine hesitancy[40]. As vaccine coverage statistics were only available for ages of 3–11 and 12–19, we used these ranges to estimate the coverage in ages 5–11 and 12–18, respectively. Misclassification bias could affect the estimates as SARS-CoV-2 infections were likely to have been underdiagnosed during the massive BA.2 wave, and this can slightly underestimate VE as there was likely more under reporting of cases and not testing for those who were unvaccinated. VE estimates of BNT162b2 and CoronaVac should not be compared, as they had different timelines of introduction and dosing intervals. Although the relatively late rollout of dose 2 CoronaVac in children partly coincided with the fall in cases that could lead to overestimation of VE[41], this estimate was consistent with our results in

adolescents and the Brazilian and Chilean studies[17,21]. Future studies using test-negative designs may reduce bias of differences in care seeking behaviour and access by vaccine status and confirm these results. We also estimated the number of infections averted by vaccination during the study period, which was higher in adolescents than children, likely because of the earlier rollout of vaccination programme and higher vaccine coverage in adolescents than in children. Our methodology likely underestimates the number of cases averted by the vaccination programme as we did not take VE against onward transmission into account, vaccination in children and adolescents may also protect other age groups, and dose 3 VE was not estimated. Furthermore, the IRR used for VE estimation differs from the risk ratio in a high-incidence setting.

In conclusion, we found preserved VE against infection in children vaccinated with BNT162b2 and CoronaVac during the BA.2 wave in Hong Kong. There have been few studies on the VE of various COVID-19 vaccine platforms in children and adolescents, and more studies should be performed to investigate VE in children and adolescents conferred by different vaccine platforms in different settings, including against future variants. Our findings support offering children and adolescents mRNA and inactivated COVID-19 vaccines.

## Data availability
Owing to data sharing restrictions by the Department of Health, individual-level data cannot be shared. The aggregate dataset for all Tables and Figures is appended in the Supplementary Data 1.

## Code availability
Standard epidemiological analyses were performed using R (Version 4.0.3). The commands are accessible via DOI:10.5281/zenodo.744490[42].

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

## Acknowledgements

The authors thank the Centre for Health Protection, Department of Health, Hong Kong Special Administrative Region Government for the provision of data. We also thank Dr Minal K. Patel for her critical review of our manuscript. This study was supported by the Providence Foundation, which was not involved in the study design, data collection, statistical computation, interpretation, or final conclusions of this project.

## Author contributions

Y.L.L. conceptualized the study. W.H.S.W., Y.L.L., D.L., and J.S.R.D. designed the study. Y.L.L. led the acquisition of data and funding. Y.L.L., and W.H.S. Wong supervised the project. W.H.S. Wong provided software support. K.M.Y., H.K.S., and W.H.S.W. curated data and estimated VE. D.L. and K.M.Y. visualized the data. D.L. wrote the first draft supervised by Y.L.L., with input from J.S.R.D., and W.H.S.W. All authors critically reviewed and approved the final manuscript.

## Competing interests

The authors declare the following competing interests: Y.L.L. chairs the Scientific Committee on Vaccine-Preventable Diseases in the Hong Kong Special Administrative Region Government. All other authors declare no competing interests.

## Additional information

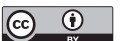

