## [Peer Review File · Communications Medicine]

This manuscript has been previously reviewed at another Nature Portfolio journal. This document only contains reviewer comments and rebuttal letters for versions considered at Communications Medicine

Reviewers' comments:

Reviewer #1 (Remarks to the Author):

Thank you for revising the manuscript. The authors have addressed the concerns brought up in my statistical review of the previous draft of the manuscript.

Reviewer #4 (Remarks to the Author):

I think that the authors have responded adequately to all of the points raised by the initial reviewers apart from point 2 of reviewer 3.

This is a point about bias associated with under reporting of cases and not testing asymptomatic individuals. The description of the system in Hong Kong is ideal but the discussion point is weak.

“Misclassification bias could affect the estimates as SARS-CoV-2 infections were likely to have been underdiagnosed during the massive BA.2 wave.”

The authors should really try to discuss the likely direction of the bias and the likely magnitude.

My comments:

Page 5 line 82/83: how did you get the confidence intervals for the numbers of cases averted? I had a look at the R code and I think that you are only taking into account variation in the IRR when calculating the confidence intervals for the cases averted. The formula you have on line 315 page 19 is a sum of random variables which are themselves ratios of random variables. Also the components of the sum are not going to be independent. You might be able to justify that the major source of the variation is the IRR and that you don't need to take into account the sampling variation in the cases. If my supposition is correct the appropriate calculation would involve the formula for the variance of a ratio using the delta method but I think that using a parametric bootstrap would be easier. Also I did not see the use of these confidence intervals anywhere in the paper.

page 11: Fig 2: why is the confidence interval for the expected number of cases so narrow for the 5-11. The VE estimates are quite wide.

Analysis of VE in 3-11. Does this analysis not suffer from the same issue as pointed out by referee 2. vaccine uptake is almost zero up to mid February, then it increases rapidly to end of March, at the same time as there is a large number of daily cases. This is just for discussion.

Point-to-point reply

Reviewer #1 (Remarks to the Author):

Thank you for revising the manuscript. The authors have addressed the concerns brought up in my statistical review of the previous draft of the manuscript.

Response: Thank you for your professional statistical review of our manuscript.

Reviewer #4 (Remarks to the Author):

1. I think that the authors have responded adequately to all of the points raised by the initial reviewers apart from point 2 of reviewer 3. This is a point about bias associated with under reporting of cases and not testing asymptomatic individuals. The description of the system in Hong Kong is ideal but the discussion point is weak. "Misclassification bias could affect the estimates as SARS-CoV-2 infections were likely to have been underdiagnosed during the massive BA.2 wave." The authors should really try to discuss the likely direction of the bias and the likely magnitude.

Response: Thank you for your professional review. We have enriched the discussion.

Line 201

"Misclassification bias could affect the estimates as SARS-CoV-2 infections were likely to have been underdiagnosed during the massive BA.2 wave, and this can slightly underestimate VE as there was likely more under reporting of cases and not testing for those who were unvaccinated."

My comments:

2. Page 5 line 82/83: how did you get the confidence intervals for the numbers of cases averted? I had a look at the R code and I think that you are only taking into account variation in the IRR when calculating the confidence intervals for the cases averted. The formula you have on line 315 page 19 is a sum of random variables which are themselves ratios of random variables. Also the components of the sum are not going to be independent. You might be able to justify that the major source of the variation is the IRR and that you don't need to take into account the sampling variation in the cases. If my supposition is correct the appropriate calculation would involve the formula for the variance of a ratio using the delta method but I think that using a parametric bootstrap would be easier. Also I did not see the use of these confidence intervals anywhere in the paper.

Response: Thank you for the above suggestion, which we followed accordingly. We have recalculated the confidence intervals for cases averted using bootstrap based on 1000 resamples and revised Fig. 2 accordingly. The CIs remain roughly similar.

Figure 2

Line 322

"95% CIs of expected cases are estimated using bootstrap based on 1000 resamples."

3. page 11: Fig 2: why is the confidence interval for the expected number of cases so narrow for the 5-11. The VE estimates are quite wide.

Response: Thank you for this insightful question by the reviewer. The expected number of cases in the absence of the vaccination programme is estimated by $case_0 + case_{BNT1}/(1-VE_{BNT1}) + case_{BNT2/3}/(1-VE_{BNT2}) + case_{CoV1}/(1-VE_{CoV1}) + case_{CoV2/3}/(1-VE_{CoV2})$, where $case_{0/BNT/CoV1/2/3}$ is the number of observed unvaccinated cases or cases with 1/2/3 doses of BNT162b2 or CoronaVac and $VE_{BNT/CoV1/2}$ is the VE of 1 or 2 doses of BNT162b2 or CoronaVac. Most of the infected cases in children aged 3-11 were unvaccinated. The numbers of observed cases in vaccinated children were low, which was the reason that the confidence intervals for the cases averted appear narrow in the graph (each minor ticks mark 200 cases, and the daily maximum of observed cases in children with 1 dose BNT162b2 were only 64 cases and 2 doses CoronaVac were only 22 cases so these are not apparent in the graphs).

4. Analysis of VE in 3-11. Does this analysis not suffer from the same issue as pointed out by referee 2. vaccine uptake is almost zero up to mid February, then it increases rapidly to end of March, at the same time as there is a large number of daily cases. This is just for discussion.

Response: Thank you for this important point brought up by the astute reviewer. We agree that there could be similar bias with regards to dose 2 CoronaVac in children aged 3-11 years, but the VE estimate for dose 2 CoronaVac in children (40.8%) was similar to adolescents (55.0%) and was consistent with VE estimates from Brazilian (42%) and Chilean (38%) studies in children as cited on line 159. Therefore, we believe that the relatively later rollout of dose 2 CoronaVac in children did not lead to overestimation of the VE. Yet, we have now noted this in the limitations.

Line 205

“Although the relatively late rollout of dose 2 CoronaVac in children partly coincided with the fall in cases that can lead to overestimation of VE,⁴⁰ this estimate is consistent with our results in adolescents and the Brazilian and Chilean studies.^{17,21”}

REVIEWERS' COMMENTS:

Reviewer #4 (Remarks to the Author):

Thanks for your responses to the points i raised. I am very content with all of your responses.